# Single-cell RNA-seq of *Drosophila miranda* testis reveals the evolution and trajectory of germline sex chromosome regulation

**Kevin H-C. Wei**[1,2]**, Kamalakar Chatla**[1]**, Doris Bachtrog**[1] *****

**1** Department of Integrative Biology, University of California Berkeley, Berkeley, California, United States of America, **2** Department of Zoology, University of British Columbia, Vancouver, British Columbia, Canada

* dbachtrog@berkeley.edu

**Data Availability Statement:** Raw reads of scRNA-seq are available on the Sequenced Read Archive under BioProject PRJNA907067. Key intermediate files including scRNA-seq mapping, orthology

## Abstract

Although sex chromosomes have evolved from autosomes, they often have unusual regulatory regimes that are sex- and cell-type-specific such as dosage compensation (DC) and meiotic sex chromosome inactivation (MSCI). The molecular mechanisms and evolutionary forces driving these unique transcriptional programs are critical for genome evolution but have been, in the case of MSCI in *Drosophila*, subject to continuous debate. Here, we take advantage of the younger sex chromosomes in *D. miranda* (XR and the neo-X) to infer how former autosomes acquire sex-chromosome-specific regulatory programs using single-cell and bulk RNA sequencing and ribosome profiling, in a comparative evolutionary context. We show that contrary to mammals and worms, the X down-regulation through germline progression is most consistent with the shutdown of DC instead of MSCI, resulting in half gene dosage at the end of meiosis for all 3 X's. Moreover, lowly expressed germline and meiotic genes on the neo-X are ancestrally lowly expressed, instead of acquired suppression after sex linkage. For the young neo-X, DC is incomplete across all tissue and cell types and this dosage imbalance is rescued by contributions from Y-linked gametologs which produce transcripts that are translated to compensate both gene and protein dosage. We find an excess of previously autosomal testis genes becoming Y-specific, showing that the neo-Y and its masculinization likely resolve sexual antagonism. Multicopy neo-sex genes are predominantly expressed during meiotic stages of spermatogenesis, consistent with their amplification being driven to interfere with mendelian segregation. Altogether, this study reveals germline regulation of evolving sex chromosomes and elucidates the consequences these unique regulatory mechanisms have on the evolution of sex chromosome architecture.

## Introduction

Sex chromosomes often show unusual expression patterns relative to autosomes. Repeat-rich and gene-poor Y chromosomes are often epigenetically silenced [1–3]. X chromosomes, in response, have evolved dosage compensation (DC) in many species to equalize expression of

calls, bulk RNA-seq read counts, and ChIP-seq peaks have been deposited on to Dryad. (https://doi.org/10.6078/D1DH7T). Analyses can be reproduced from S1 Data, which includes all necessary files and R codes to generate the figures and underlying data.

**Funding:** DB received funding from the National Institutes of Health (https://www.nih.gov/) (5R01GM101255). The funders played no role in study design, data collection, analysis, decision to publish or preparation of the manuscript.

**Competing interests:** The authors have declared that no competing interests exist.

**Abbreviations:** CES, chromatin entry site; DC, dosage compensation; GSC, germline stem cell; MSCI, meiotic sex chromosome inactivation; MSL, male-specific lethal; SC, spermatocytes; scRNA-seq, single-cell RNA-seq; SG, spermatogonia; smRNA, small RNA; ST, spermatids.

the sex chromosomes and autosomes in both sexes [4–7]. This is accomplished via diverse mechanisms found in different taxa including down-regulation of 1 copy of the X in female mammals and reduced expression of both X copies in worms [5,8,9]. In *Drosophila*, DC occurs via hyper transcription of the single X in males. This is achieved via the sequence-specific recruitment of the male-specific lethal (MSL) complex to so-called chromatin entry sites (CESs) on the X which induces the activating histone tail modification H4K16 acetylation (H4K16ac) at nearby transcribed genes [10–12].

Germline regulation of sex chromosomes also appears to be uniquely distinct from autosomes. In many organisms, expression on the X is highly reduced during meiosis of spermatogenesis, a phenomenon known as meiotic sex chromosome inactivation (MSCI) [13–15]. In mammals, the X is sequestered in a silencing compartment in the nuclear periphery known as the sex-body during late meiotic prophase [16–20]. Similarly, the germline of XO male worms creates a structure analogous to the sex-body, where the X is associated with the repressive histone tail modification H3K9-methylation [13,21]. Despite the apparent conservation, the evolutionary importance of MSCI remains unclear but MSCI may prevent unwanted recombination between the heterologous sex chromosomes [22], safeguard the meiotic progression of asynapsed chromosomes [23], silence the expression of a feminized X chromosome to reduce sexual antagonism [24] or inactivate sex-linked selfish genetic elements that distort sex ratio [25–27].

Interestingly, unlike other model organisms such as mice and worms, the status of MSCI in the male germline of *Drosophila* has been contentious [28–36]. In the apical tip of the fly testes, germline stem cells (GSCs) are supported by somatic hub and cyst cells. After differentiation, the spermatogonia (SG) undergo typically 4 rounds of mitotic divisions with incomplete cytokinesis resulting in multinucleated cysts containing 16 spermatocytes (SC) that enter meiotic division. Late spermatocytes exit meiosis forming haploid spermatids (ST) that remain connected until spermiogenesis when sperm maturation, individualization, and elongation occur to form mature sperm bundles. X-linked expression across the germline appears to be reduced when assayed with qPCR of targeted genes [30,34], reporter constructs [28,34], and chromosome-wide transcriptome analyses [29–32]. However, whether the regulatory basis of this down-regulation is specific to meiotic tissues or can be attributed to MSCI has been debated. In particular, the status of dosage compensation confounds the interpretation of germline X-regulation in the *Drosophila* testes. While MSL-dependent dosage compensation is well-characterized in somatic tissues, its presence in the testes is unclear. Cytological studies of testis have failed to detect the MSL complex or the H4K16ac modification it induces in GSC, SG, and SC [37]. However, recent genomic studies have revealed X-linked expression patterns and nuclear topologies which suggest that a distinct mechanism maintains X dosage in the premeiotic stages of spermatogenesis [20,35,38].

Gene content and gene expression evolution further complicates inferences of chromosome-wide expression patterns of sex chromosomes during spermatogenesis. Genes that are expressed in testis have been found to move off the X chromosome [39–42], while sexually antagonistic selection could result in an excess or deficiency of genes with male- or female-biased expression on sex chromosomes, depending on underlying population parameters [43–46]. In particular, female-biased transmission of the X may select for an excess of female-specific genes/a deficiency of male-specific genes, and overall lower expression of the X compared to autosomes in males ("demasculinization" of the X; [43,47]). Male-beneficial genes, on the other hand, should be overrepresented on the male-specific Y chromosome [43,48–50].

Because sex chromosomes typically have autosomal origins [51,52], newly sex-linked chromosomes will gradually acquire regulatory features of older sex chromosomes over evolutionary time. Due to its unusual sex chromosome architecture, *Drosophila miranda* has been a model species for understanding sex chromosome evolution (**Fig 1A**). Unlike the ancestral

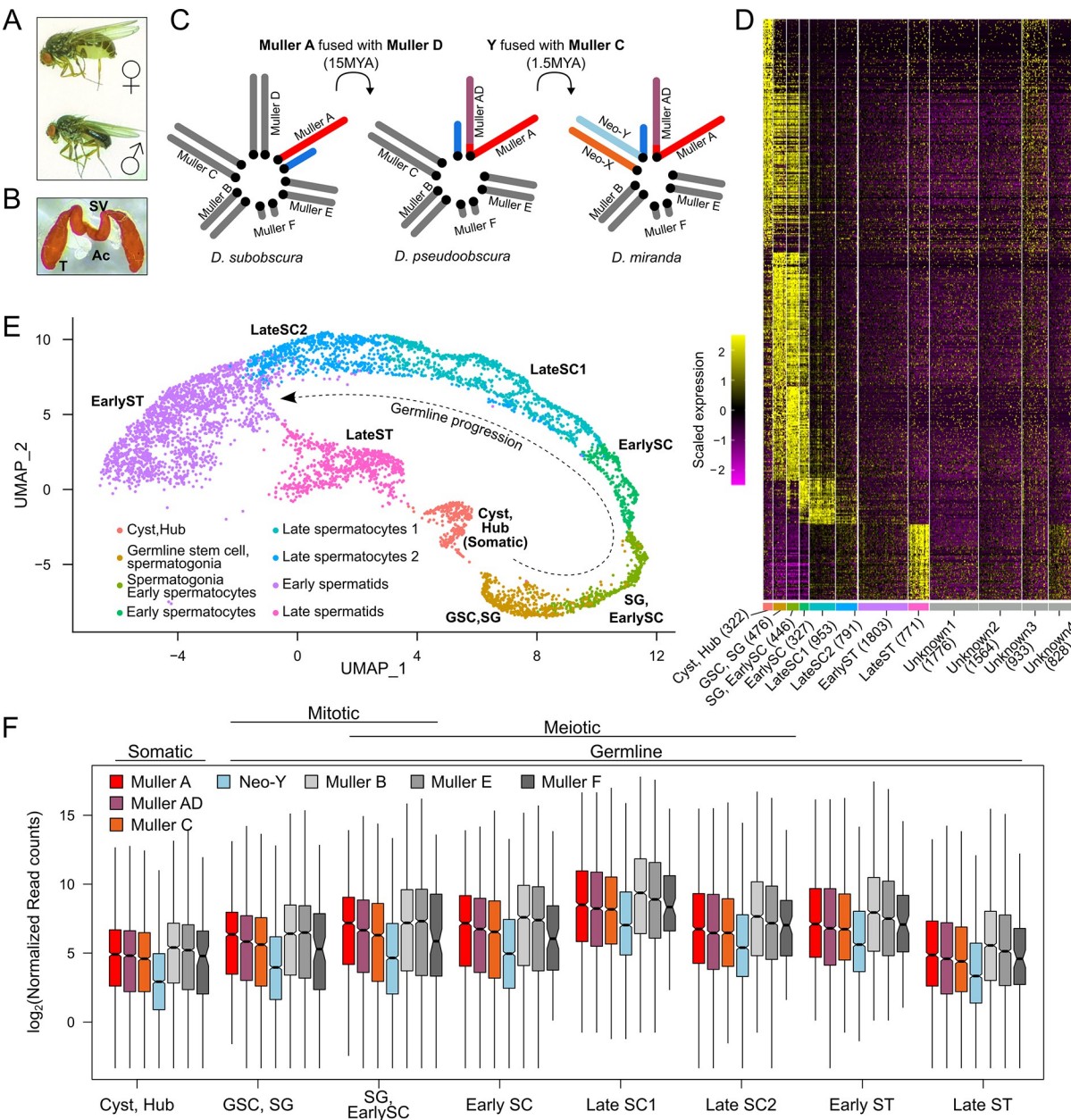

**Fig 1. Transcriptional dynamic of *D. miranda* testes at single-cell resolution.** (**A**) Picture of *D. miranda* female and male adults. (**B**) Picture of a pair of testes (T) attached by the seminal vesicle (SV) to the accessory glands (Ac). (**C**) Male karyotype evolution showing multiple fusions creating neo-sex chromosomes of different ages. Red, purple, and orange denote the 3 X-chromosomes of different ages: Muller A (~50MY), Muller AD (~15MY), and Muller C (~1.5MY), respectively. The Y chromosome arms are in shades of blue, and the autosomes are in gray. Unless otherwise stated, this color-coding of the chromosomes will be maintained throughout. (**D**) Expression of genes used for clustering cells into cell types; number of cells (UMIs) in each type is in parentheses. Yellow and magenta represent high and low expression. (**E**) UMAP reduced dimension projection showing cells from scRNA-seq clustered into 8 different cell types corresponding to different stages of spermatogenesis. Progression of spermatogenesis is outlined by curved arrow. For projection of all cells including unidentified clusters, see S1 Fig. (**F**) Distributions of normalized gene counts from different chromosomes across cell types. For each gene, counts are aggregated across cells of the same cell types. Boxplots depict the distribution of normalized expression for each chromosome and cell type. Cell types are labeled as somatic, germline, mitotic, and meiotic above the plot. For number of reads in each cell type and chromosome, see S1 Table. For number of genes used for each chromosome, see S2 Table. The data underlying this figure can be found in S1 Data.

karyotype of *Drosophila* where a single chromosome arm (Muller A) is the X chromosome, *D. miranda* has 2 additional X chromosome arms that are younger (**Fig 1C**). A metacentric fusion linked one autosome, Muller D, with Muller A ~15MYA, causing the former to become X-linked. Muller D has evolved the stereotypical characteristics of a sex chromosome, with the non-recombining homolog almost fully degenerate, and the X-linked homolog having evolved full dosage compensation [53] (note that we refer to this chromosome arm as Muller AD, since a pericentric inversion moved some Muller A genes onto Muller D). A secondary fusion between Muller C and the Y occurred around ~1.5MYA creating a Y-linked copy of Muller C that is male exclusive (i.e., the neo-Y). Its former homolog, while not fused to Muller A and D, is transmitted as 2 copies in females and 1 copy in males, making it a neo-X chromosome. The neo-X and neo-Y, once a homologous pair of autosomes, have been evolving characteristic traits of sex chromosomes, like X dosage compensation [2,54] and Y degeneration [55–58], and are at an intermediate stage in the transition from an ordinary autosome to a pair of differentiated sex chromosomes [57,58]. The acquisition of dosage compensation on the neo-X occurred through co-opting insertions of a transposable element that harbors the sequence motif targeted by the MSL complex [53,59].

D. miranda's unique sex chromosome architecture offers an ideal opportunity to characterize meiotic sex chromosome regulation. Specifically, because the evolving neo-X represents an intermediate to autosomes and the X, the status of MSCI can be tested without the confound of significant demasculinization. Further, because the neo-X is a separate chromosome from Muller A and AD, X-specific regulation must arise independently on this chromosome and cannot simply be explained by linkage to the older X's. To this end, we employed single-cell RNA-seq (scRNA-seq) on *D. miranda* testes (**Fig 1C**) to disentangle the regulatory mechanisms of sex chromosomes during spermatogenesis and to understand how young sex chromosomes evolve in response to such meiotic regulatory regimes. We find expression patterns consistent with incomplete premeiotic dosage compensation of the neo-X followed by gradual loss of dosage compensation of all X's across meiotic progression. Our results suggest that shutdown of dosage compensation is the primary source of meiotic down-regulation of the X's. By comparing expression patterns of the neo-X to its autosomal ortholog in *D. pseudoobscura* using bulk tissue RNA-seq, we show that low expression of a subset of genes on the neo-X reflects ancestrally low expression levels rather than being explained by MSCI. Curiously, incomplete dosage compensation causes reduced neo-X expression across all tissues which is further exacerbated in testes due to postmeiotic DC shutdown. This expression deficit appears to be compensated by neo-Y gametologs, despite extensive degeneration and down-regulation of the neo-Y chromosome. Moreover, previously autosomal genes with testes-biased expression are disproportionally lost from the neo-X, thus becoming exclusively Y-linked, likely to resolve sexual antagonism. Interestingly, these now Y-exclusive genes are primarily expressed late or after meiosis, raising the possibility that postmeiotic germline regulation may be a major source of sexual antagonism driving the genetic architecture of sex chromosomes. Moreover, we show that co-amplified neo-sex genes are predominantly expressed during meiotic stages of spermatogenesis, consistent with their amplification being driven to compete for transmission to the next generation. In summary, our results shed light on the various molecular and selective processes operating on sex chromosomes and how they evolve.

## Results

### Transcriptional landscape of germline progression at single-cell resolution

We generated scRNA-seq libraries from dissociated *D. miranda* testes cell preparations in duplicates. After processing and filtering, the replicated scRNA-seq experiment yielded 10,990

cells that fall into 12 clusters (**Figs 1D** and **S1A**). We identified the cell types of 8 clusters using testes marker genes characterized in *D. melanogaster* (**Figs 1E** and **S1B**). Four clusters show low but similar expression to the spermatid clusters but lack obvious expression profiles of distinguishing marker genes, thus we removed them from our downstream analyses (**S1B Fig**). This results in 8 cell types totaling 5,101 cells that span spermatogenesis, with germline progression roughly following a linear trajectory in the UMAP projection (**Fig 1E** and **S1 Table**).

Spermatogenesis in *D. miranda* and species in the *obscura* group has several unique features that differ from *D. melanogaster*. First, spermatogonia undergo 5 rounds of mitotic divisions (instead of 4), thereby producing 32-nuclei (instead of 16) spermatocytes. Second, flies from the *obscura* group produce heteromorphic sperm types (the fertilizing eusperm and sterile parasperm) [60,61]. While these physiological differences likely contribute to the different numbers of cells and cell type clusters compared to previous studies of *D. melanogaster* testes [35,62], the clustering results are largely consistent. Overall transcript abundance across all chromosomes increases through germline progression (**Figs 1F** and **S2**) and peaks in late spermatocyte cell types; many genes (including cup and comet genes [63]) are up-regulated in preparation for sperm individualization and maturation (**S1B Fig**). We further noticed that the spermatocyte stages display 2 subpopulations (streaks in the UMAP projections) which can be separated using more sensitive clustering (**S3 Fig**). They may represent the developmental trajectories of the 2 sperm types which are produced in equal proportions in *D. miranda* [64]. However, given that the genetic basis of the heteromorphic sperm types is currently unknown, we were unable to confirm or assign sperm types to these subclusters.

Comparing between expressed genes on the chromosomes (**S2 Table**), we find that the X chromosome arms consistently show lower expression than the autosomes (except for Muller F, which used to be an X chromosome [52,65]; **Fig 1F**). This is true even in the somatic tissue where the X is expected to be fully dosage compensated. Lower expression from the X in male tissues compared to female tissues has been interpreted as an adaptation of the X to female-biased transmission (i.e., feminization of the X).

Nevertheless, across the stages, the ancestral X, Muller A, typically has the highest expression while the neo-X has the lowest expression. Because the neo-X, Muller C, has had little time to demasculinize, the low expression is more likely due to incomplete dosage compensation of the neo-X (see below). Consistent with the neo-Y degenerating and becoming heterochromatic, genes on the neo-Y are lowly expressed compared to all other chromosomes. Expression of all chromosomes peaks at late spermatocytes (**Fig 1F**); however, genes from the neo-Y have the largest fold-increase between early and late spermatocyte stages, revealing that the neo-Y is disproportionally up-regulated in late meiosis ($p < 0.00001$, Wilcoxon's rank sum test; **S2 Fig**) [66].

## Reduced expression of X linked genes through meiosis consistent with shutdown of dosage compensation

*D. miranda's* unique sex chromosome architecture presents an opportunity to test the status of X chromosome regulation in the germline and to understand its evolution (**Fig 2A**). If meiotic X-inactivation is an evolved trait in the male germline, the neo-X—which is at an intermediate stage of evolving X characteristics—should show expression patterns still reminiscent of its recent autosomal history. That is, genes on the neo-X will be expressed at a level that is intermediate of autosomal (not inactivated) and X (inactivated) levels during meiosis. Alternatively, given the presence of dosage compensation in the early germline [35], reduced meiotic expression may reflect shutdown of dosage compensation post meiosis instead of targeted inactivation. Since the neo-X is still evolving towards becoming fully dosage compensated, loss of

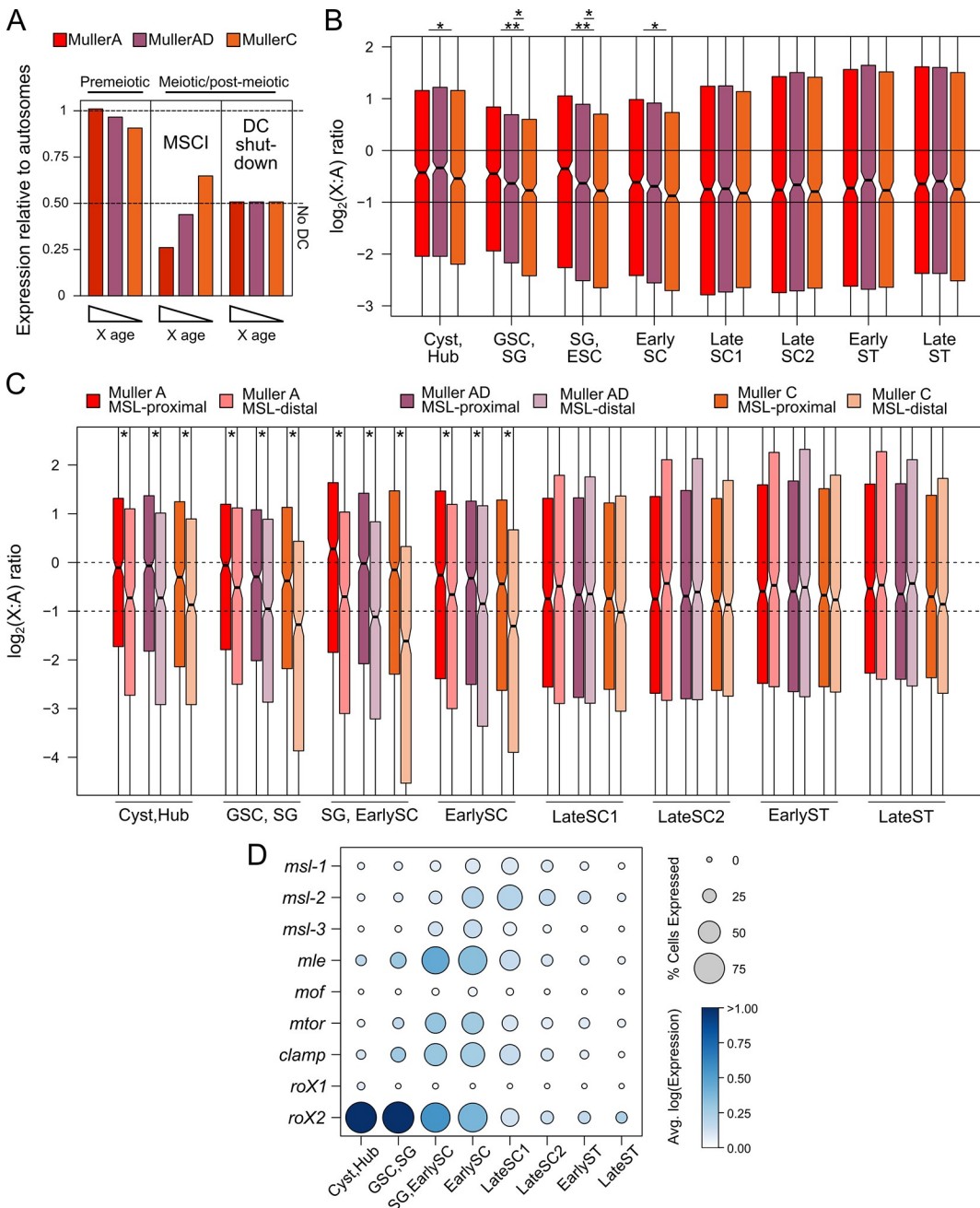

**Fig 2. Gradual loss of dosage compensation through meiosis. (A)** Hypothetical models of MSCI vs. loss of dosage compensation in Xs with different ages. **(B)** Distribution of X-to-autosome ratios across cell types. * = $p < 0.001$ and ** = $p < 0.000001$, Wilcoxon's rank sum test. Boxplots represent the distribution of X-linked gene expression divided by the median expression of autosomal genes after aggregating read counts across cells of the same cell type. **(C)** X-to-autosome ratios of genes based on their proximity to MSL-binding sites. Genes <2 kb or >2 kb away are classified as proximal or distal, respectively. **(D)** Expression of the genes in or associated with the MSL complex across cell clusters. Size of the circle is proportional to the fraction of cells in the cluster with expression of the gene and the intensity of the color reflects the expression in log scale averaged across cells within the cluster. The data underlying this figure can be found in S1 Data. MSCI, meiotic sex chromosome inactivation; MSL, male-specific lethal.

dosage compensation predicts a distinctly different pattern from MSCI. Under this scenario, the neo-X would show lower expression compared to the other X arms prior to meiosis and through meiosis all 3 X chromosomes will lower to similar expression levels as dosage compensation attenuates (**Fig 2A**).

To differentiate between these scenarios, we contrasted the expression of the 3 X's relative to the autosomes throughout spermatogenesis (**Fig 2B**). We find that prior to meiosis (in GSC, SG, and SC), the neo-X consistently has the lowest expression, with Muller AD showing intermediate expression levels between the ancestral X and neo-X (**Fig 2B**). This stepwise difference in expression among X chromosomes during spermatogenesis is consistent with the gradual acquisition of dosage compensation over evolutionary time, where the ancestral X is fully dosage compensated, the neo-X shows partial dosage compensation, and Muller AD is in between (**Fig 2B**). After meiosis (in late SC and early ST), all 3 X chromosomes are becoming consistently lower expressed than autosomes (**Fig 2B**). In particular, expression levels of the older X chromosome arms begin to more closely resemble that of the neo-X across meiotic progression, equalizing at meiotic exit (late SC1). Most importantly, expression of all the X's never drops below half that of the autosomes. This pattern, which can also be observed when using only testes-expressed genes (**S4 Fig**), supports the loss of dosage compensation as the mechanism underlying reduced X expression in testis and is inconsistent with the prediction that the neo-X is evolving X-inactivation. Altogether, our results argue that the X's are not inactivated or severely down-regulated through meiosis but are fully consistent with shutdown of dosage compensation during meiosis.

## Loss of dosage compensation for genes proximal to MSL-bound regions across meiosis

To further evaluate the loss of dosage compensation across meiosis, we identified regions bound by the dosage compensation complex using larval ChIP-seq data [2,54] targeting the MSL3 protein (a component of the MSL complex). We identified 1513, 2762, and 2115 MSL-bound regions for Muller A, AD, and C, respectively, and then evaluated expression differences between genes proximal ($<2$ kb) and distal ($>2$ kb) to MSL-bound sites. Consistent with the MSL-complex inducing transcriptional up-regulation of X-linked genes, genes proximal to MSL-bound regions show significantly higher expression on all 3 X chromosomes in the somatic tissues and prior to meiosis (**Fig 2C**; Wilcoxon's rank sum tests, $p < 0.0001$). However, as meiosis progresses and dosage compensation becomes lost, the expression of the 2 sets of genes becomes more similar and upon meiotic exit becomes statistically indistinguishable in most comparisons (**Fig 2C**; Wilcoxon's rank sum test, $p > 0.1$). Thus, these patterns are again consistent with reduced expression of X-linked genes during spermatogenesis in *Drosophila* resulting from a shutdown of dosage compensation. Associations of expression patterns for X-linked genes with MSL-complex binding (although inferred from larvae) raises the possibility that (members of) the MSL-complex also have a role for dosage compensation in testis (but see [37]).

To specifically examine the activity of the dosage compensation complex, we looked at the expression of components of the MSL complex and 2 accessory genes that are important for dosage compensation (*clamp* and *mtor*, **Fig 2D**). With the exception of *mof* and *roX1*, which have low expression across all cell stages (including the somatic cyst and hub cells and premeiotic stages), transcript levels of genes important for dosage compensation are intermediate-to-high in the premeiotic and early meiotic cell types. The long noncoding RNA, *roX2*, has the highest expression in the somatic cells and GSC, and gradually diminishes across germline progression. The others increase in expression peaking at the spermatocyte stages and decrease

after. These expression patterns indicate that most genes involved in dosage compensation are active during early spermatogenesis and are gradually lost through meiosis, consistent with dosage compensation shutdown.

## Distinct germline transcriptional trajectories depending on MSL proximity

Consistent with DC shutdown, the expression differences between MSL-proximal and distal genes become smaller during later spermatogenesis (**Fig 2C**). Curiously, in the early germline (SG and early SC), MSL-proximal and distal genes show the greatest expression difference, and the distal genes are expressed drastically lower than half of autosomal expression. To better understand these expression changes, we refined our classification of X-linked genes into those overlapping MSL peaks, within 1 kb, 5 kb, 20 kb, and >20 kb (**Figs 3A and S5A**). Regardless of the cell types, MSL-proximity is consistently associated with higher expression (**S5 Fig**). Genes farthest away from MSL sites (>20 kb) present the only exception and are almost always more highly expressed than genes between 5 and 20 kb of MSL sites—this is consistent with male-biased genes being depleted near dosage compensation sites [67]. The association between MSL-proximity and gene expression is greatest in the early germline stages and reduces in the later stages consistent with the equalizing effect of DC shutdown. However, higher expression of genes near MSL sites remains, even after DC shutdown, likely because MSL sites are more common near highly expressed genes (see below).

Despite becoming more similar in expression on average, MSL-distal and -proximal genes show distinct regulatory trajectories across germline progression (**Fig 3A**). Genes near MSL sites are first up-regulated in early germline development followed by gradual decrease through meiosis, consistent with DC shutdown. The early up-regulation of MSL-proximal genes occurs extensively as the SGs progress into SCs when multiple rounds of mitosis occur. At the highest, genes overlapping MSL peaks reach elevated X:A ratio of 1.98, 1.36, 1.63, and for Muller-A, AD, and C, respectively (**S5A Fig**). This pattern of early up-regulation was also observed in *D. melanogaster* and interpreted as non-canonical hyper dosage compensation [35]. Whereas all genes within 5 kb of MSL peaks show early up-regulation on the older X arms (Muller A and AD), up-regulation on the neo-X (Muller C) is restricted to only genes overlapping MSL peaks. This is consistent with the suboptimal nature of recently evolved DC on the neo-X resulting in less MSL-dependent spreading of the activating chromatin environment.

For MSL-distal genes, we observed the opposite pattern whereby they are lowly expressed during the early stages, followed by dramatic up-regulation through meiotic stages (**Fig 3A**). This early down-regulation results in extremely low expression of the MSL-distal genes upon meiotic entry, well-outside the bound of the absence of DC, with median X:A ratios of 0.259, 0.129, and 0.069, for Muller-A, AD, and C, respectively (**S5A Fig**). Upon meiosis, the MSL-distal genes begin increasing in expression with the most MSL distal showing the greatest up-regulation. Further examination revealed that this meiotic increase is driven by up-regulation of a large subset of MSL-distal genes, while the rest remain lowly expressed, resulting in a bimodal distribution of gene expression for all 3 X chromosome arms (**S6 Fig**).

## Low meiotic expression of a subset of neo-X genes reflects ancestral expression

The MSL complex has been shown to target actively expressed genes by recognizing the H3K36me3 histone mark, which is disproportionally found on house-keeping genes [68,69]. As a consequence, cell type-specific genes such as those expressed only during and after meiosis—at a time when DC no longer is active—are naturally going to be far from MSL.

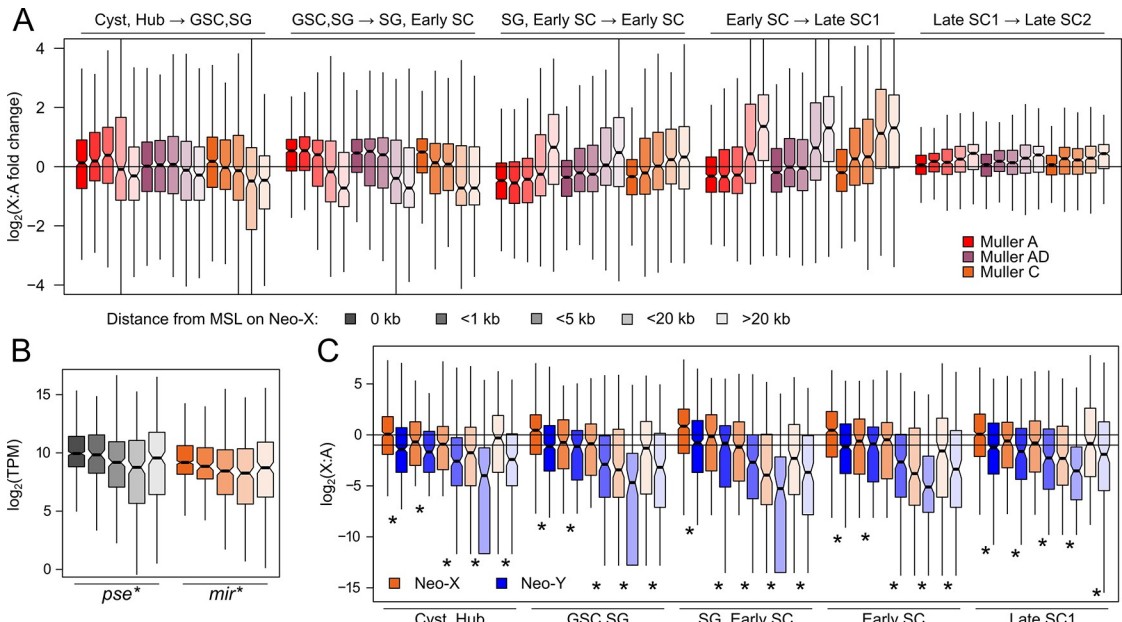

**Fig 3. MSL-distance associated with distinct germline regulation and ancestral expression patterns. (A)** Genes on the X chromosomes are subcategorized into 5 groups depending on distance to MSL peaks (for number of genes in each category, see S3 Table). The fold changes in X:A ratio between testes cell stages are depicted in log scale for the different gene categories, with greater and less than 0 values representing increase and decrease in expression. **(B)** Autosomal (*D. pseudoobscura*) vs. X-linked (*D. miranda*) Muller C expression from bulk testes RNA-seq depending on proximity of the neo-X alleles to MSL sites in *D. miranda*. * = $p < 0.00001$ for Pearson's correlation between MSL distance and gene expression. **(C)** Expression of Muller C gametologs on the neo-X and neo-Y across relevant meiotic stages depending on the proximity of the neo-X alleles to MSL sites; * = *p*-value < 0.0001, Wilcoxon's rank sum test. The data underlying this figure can be found in S1 Data. GSC, germline stem cell; MSL, male-specific lethal.

Nevertheless, one could argue that low expression of MSL-distal genes upon meiotic entry is the result of MSCI. To evaluate this possibility, we examined their autosomal orthologs in *D. pseudoobscura*. If the low expression is due to MSCI, we expect these MSL-distal genes on the neo-X to have higher expression in *D. pseudoobscura*, where Muller C is autosomal. In bulk testes RNA-seq, we found that the expression of the autosomal orthologs, despite not having DC, is similarly associated with the proximity to MSL peaks in *D. miranda* (Fig 3B); i.e., distance to MSL peaks in *D. miranda* correlates with the expression in *D. pseudoobscura*. The *D. miranda* neo-X alleles are on average approximately 0.594-fold lower expressed than their autosomal counter parts, a value consistent with a mixture of cells with partial DC and DC shutdown (S7 Fig). Moreover, MSL-distal genes do not show more extensive down-regulation in *D. miranda*, as would be predicted if MSCI silences MSL-distant genes on the neo-X. These results strongly argue that low expression of MSL-distal genes is not an acquired feature of the neo-X, but simply a product of ancestral expression patterns and DC targeting highly expressed genes.

A limitation of the above analysis is the use of bulk testis expression data. This can be overcome by using neo-Y gametolog expression as a proxy for ancestral expression levels of neo-X genes on a single-cell level. If neo-X genes are silenced in specific cell types by MSCI, we would expect neo-Y gametologs to have higher expression. Contrary to this prediction, neo-Y alleles consistently show significantly lower expression across all cell stages, reflecting wide-spread Y degeneration instead of X silencing (Fig 3C). Further, the neo-X gametologs' proximity to MSL similarly predicts the expression of the neo-Y gametologs whereby MSL-proximal and

distal genes on the neo-X also have highly and lowly expressed neo-Y gametologs, respectively. This again suggests that low expression of MSL-distal genes on the neo-X (Muller C) reflects low ancestral expression, similar to our analysis using *D. pseudoobscura* autosomal orthologs.

Furthermore, the drastically reduced X:A ratio of MSL-distal genes during early germline stages is primarily caused by increases in the autosomal expression (**Fig 1F**), thus inflating the denominator and driving down the X:A ratio. Indeed, the absolute expression levels of these lowly expressed MSL-distal genes remain similar across the early meiotic stages (**S5B Fig**).

## Incomplete dosage compensation of the neo-X is exacerbated in the germline

Although our data show that all 3 X chromosomes of *D. miranda* are dosage compensated early in the male germline, the neo-X is consistently expressed at a lower level relative to the older X chromosomes, suggesting that dosage compensation on the neo-X may be incomplete. To specifically determine the extent to which the neo-X is dosage compensated in *D. miranda* males, we compared expression of orthologs between *D. miranda* and its sister species *D. pseudoobscura* where this chromosome (Muller C) remains autosomal (**Fig 4A**). We used bulk RNA-seq data from whole testis and other sexed developmental stages and tissues including embryonic stages, larvae, head, and carcass from the 2 species. While genes on Muller A and Muller AD (X-linked chromosome arms in both species) show similar expression values in the 2 species (**Fig 4B**, top and middle), the neo-X is under-expressed in several tissues of *D. miranda* males relative to Muller C in *D. pseudoobscura* (**Fig 4B**, bottom). Prior to zygotic genome activation, the embryo contains predominantly maternally deposited mRNA. Expectedly, in embryonic stage 2 (prior to zygotic genome activation in *Drosophila*), the difference between *D. pseudoobscura* and *D. miranda* expression of Muller C genes shows minimal bias with a median fold difference of 1 (**Fig 4B**, bottom). As zygotic expression ramps up through early embryonic development, neo-X/Muller C genes become increasingly *D. pseudoobscura*-biased in expression and stabilize in adult somatic tissues where the *D. miranda* alleles are expressed on average 0.769-fold less than the *D. pseudoobscura* alleles (**Fig 4B**). Therefore, neo-X linked genes are not fully dosage compensated across somatic cell lineages. In contrast, expression of Muller C genes in female tissues is highly similar between the 2 species, and the largest difference is a mere 0.1-fold reduction in *D. miranda* early embryos (**S8 Fig**).

As expected from the loss of dosage compensation through spermatogenesis (**Figs 2 and 3**), testes show the most striking difference between the species. Here, *D. miranda* orthologs are expressed on average 0.565-fold of their *D. pseudoobscura* counterparts, a reduction that is significantly greater than any other tissue (**Fig 4B**, bottom, Wilcoxon rank sum tests, $p < 10e\text{-}8$). However, while reduced expression of neo-X-linked orthologs is most extreme in the testes, the reduction is close to but nonetheless within 2-fold. Therefore, our results comparing the orthologs are consistent with the loss of dosage compensation across meiosis causing neo-X linked genes in some of the testes cell populations to have half of the expression of their autosomal counterparts in *D. pseudoobscura*.

Expression differences between orthologous genes proximal and distal to MSL-bound sites (in *D. miranda*) further confirm incomplete dosage compensation of the neo-X in *D. miranda* males, regardless of tissue type. Neo-X-linked genes distal to MSL-bound sites tend to be significantly more biased towards the *D. pseudoobscura* ortholog across most somatic tissues (**Fig 4C**; Wilcoxon's rank sum tests, $p < 0.01$), consistent with reduced or a lack of dosage compensation. However, while proximity to MSL sites reduces the *D. pseudoobscura*-bias consistent with dosage compensation, MSL-proximal genes on the neo-X are still expressed less than their autosomal orthologs. The disparity between MSL-proximal and distal genes is most

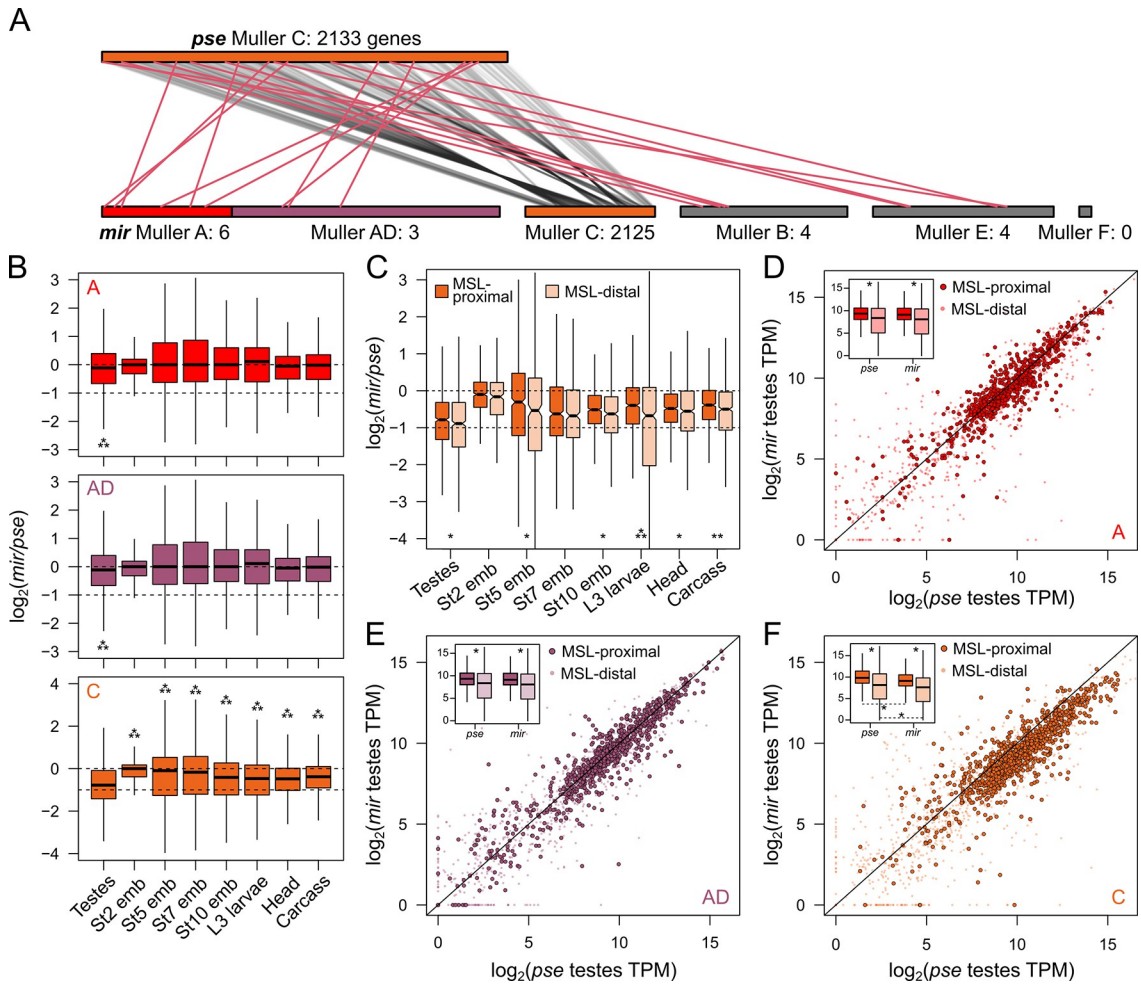

**Fig 4. Divergence of X-linked expression. (A)** Gene movement and chromosomal positions of orthologs on Muller C in *D. pseudoobscura* before and after the formation of the neo-X in *D. miranda*. Orthologous genes are connected by lines. Genes that migrated away (inter-chromosomal movement) from Muller C in *D. miranda* are in red. Number of genes from Muller C is noted next to the chromosome. Note, Muller C of *D. pseudoobscura* is drawn 4× the size for clarity. **(B)** Log2(fold-difference) between *D. pseudoobscura* and *D. miranda* orthologs on different X-chromosomes and across different tissues and developmental stages. Note, the Y-axis differs on the bottom panel corresponding to Muller C. *** = $p < 0.000001$, Wilcoxon's rank sum test. **(C)** Same as B, but Muller C genes are categorized based on their proximity to MSL-binding sites. * = $p < 0.01$, ** = $p < 0.0001$, and *** $p < 0.000001$. **(D–F)** Correlation of orthologs expression on different X-chromosomes between *D. pseudoobscura* and *D. miranda*. Genes proximal to MSL-binding sites (<2 kb) are displayed as darker circles with black borders. Diagonal line demarcates the identity line. The insets boxplots display the TPM distribution of MSL proximal and distal genes in the 2 species. * = $p < 0.0000001$. The data underlying this figure can be found in S1 Data. MSL, male-specific lethal.

drastic in the L3 larvae, likely because the MSL-chip data were collected in the same developmental stage. However, even in L3 larvae, the MSL-proximal alleles are still on average 0.76-fold lower than their *D. pseudoobscura* orthologs. Therefore, even with wide-spread recruitment of MSL on the neo-X, dosage compensation is incomplete.

For the bulk testis samples, the difference between the MSL-proximal and distal genes are marginal but significant with average fold differences of 0.580 and 0.540, respectively (Wilcoxon's rank sum test, $p < 0.01$). This slight difference is consistent with the loss of dosage compensation in many testis-derived cells (due to dosage compensation being abolished during meiosis). Looking across all 3 X chromosome arms (**Fig 4D–4F**), we find that genes near MSL-bound sites are enriched for highly expressed genes in both species. For the neo-X/Muller C

(**Fig 4F**), we find that the *D. miranda* orthologs are consistently expressed less in the testes compared to *D. pseudoobscura*, regardless of their proximity to MSL-binding sites. Therefore, despite the presence of dosage compensation in some testes cell types, genes on the neo-X remain heavily under-expressed.

## Neo-Y gametologs mitigate neo-X dosage imbalance

Because of incomplete dosage compensation across tissues and its shutdown during spermatogenesis, the neo-X of *D. miranda* appears to be expressed suboptimally. How can males maintain high fitness in light of this pervasive dosage imbalance? To address this question, we examined gene expression of the neo-Y. This chromosome shows extensive degeneration of protein-coding genes, repeat accumulation and heterochromatinization, but still harbors thousands of genes (gametologs) previously found on Muller C. Note that these gametologs are generally under reduced purifying selection and harbor an excess of deleterious amino acid mutations, and many contain premature stop codons or frame shift mutations. We identified gametologs on the neo-sex chromosomes and found that 1,692 genes maintain both neo-Y and neo-X gametologs, 339 neo-X-linked genes with no neo-Y gametologs (neo-X-only), and only 36 neo-Y-linked genes with no neo-X gametologs (neo-Y-only) (**Fig 5A**). As expected for a heterogametic chromosome that is degenerating and not dosage compensated, most genes maintained on both neo-sex chromosomes are down-regulated on the neo-Y compared to their autosomal orthologs in *D. pseudoobscura* (**S9 Fig**) and their neo-X counterparts (**Fig 5B**).

However, despite overall down-regulation, many neo-Y gametologs are still highly expressed (**Fig 5B**). This raises the possibility that neo-Y linked expression may mitigate dosage imbalance from incomplete dosage compensation, or its loss in the testes. Indeed, when we combined the expression of the neo-X and neo-Y gametologs, expression of Muller C genes in *D. miranda* testes much more closely matches autosomal expression of their orthologs in *D. pseudoobscura*, resulting in a difference of only 0.92-fold (**Fig 5C–5E**). In fact, dosage mitigation is not specific to testes, as dosage is generally restored to autosomal levels across all somatic tissues when expression from both the neo-X and neo-Y are combined (**Fig 5D**). This suggests that expression from the neo-Y linked gametologs ensures proper transcript dosage when dosage compensation is suboptimal or not fully evolved.

However, expression of neo-Y alleles alone may not be sufficient to restore dosage at the protein level, since many Y-linked alleles have accumulated premature stop codons and may not be translated into proteins. We therefore further evaluated translation of these transcripts using ribosome profiling of larval samples (**Fig 5F**). Because premature stop codons lead to rapid degradation of mRNA via the nonsense mediated decay pathway [70,71], we expect nonfunctional transcripts from the Y to have low contribution to the translating mRNA. While neo-Y gametologs with truncated CDSs have significantly reduced ribosome profiling reads (**S10A Fig**), we nevertheless find many with high ribosome occupancy suggesting they are actively translated (**Fig 5F**). Unsurprisingly, while amounts of translating neo-Y transcripts are significantly lower than all other chromosomes (**Figs 5F and S10B**), they supplement corresponding neo-X gametologs, thereby increasing the total translation rate of Muller C genes to near autosomal levels. Thus, with incomplete dosage compensation of the neo-X, the neo-Y provides an additional source of transcript and downstream protein production.

## Neo-Y as an environment to resolve sexual conflict

Despite overall reduction in expression and gene loss, the neo-Y harbors 36 genes where the neo-X gametologs are lost. These neo-Y-only genes represent a set of genes that have become male-exclusive. Unlike the bulk of neo-Y linked genes that have lower expression compared to

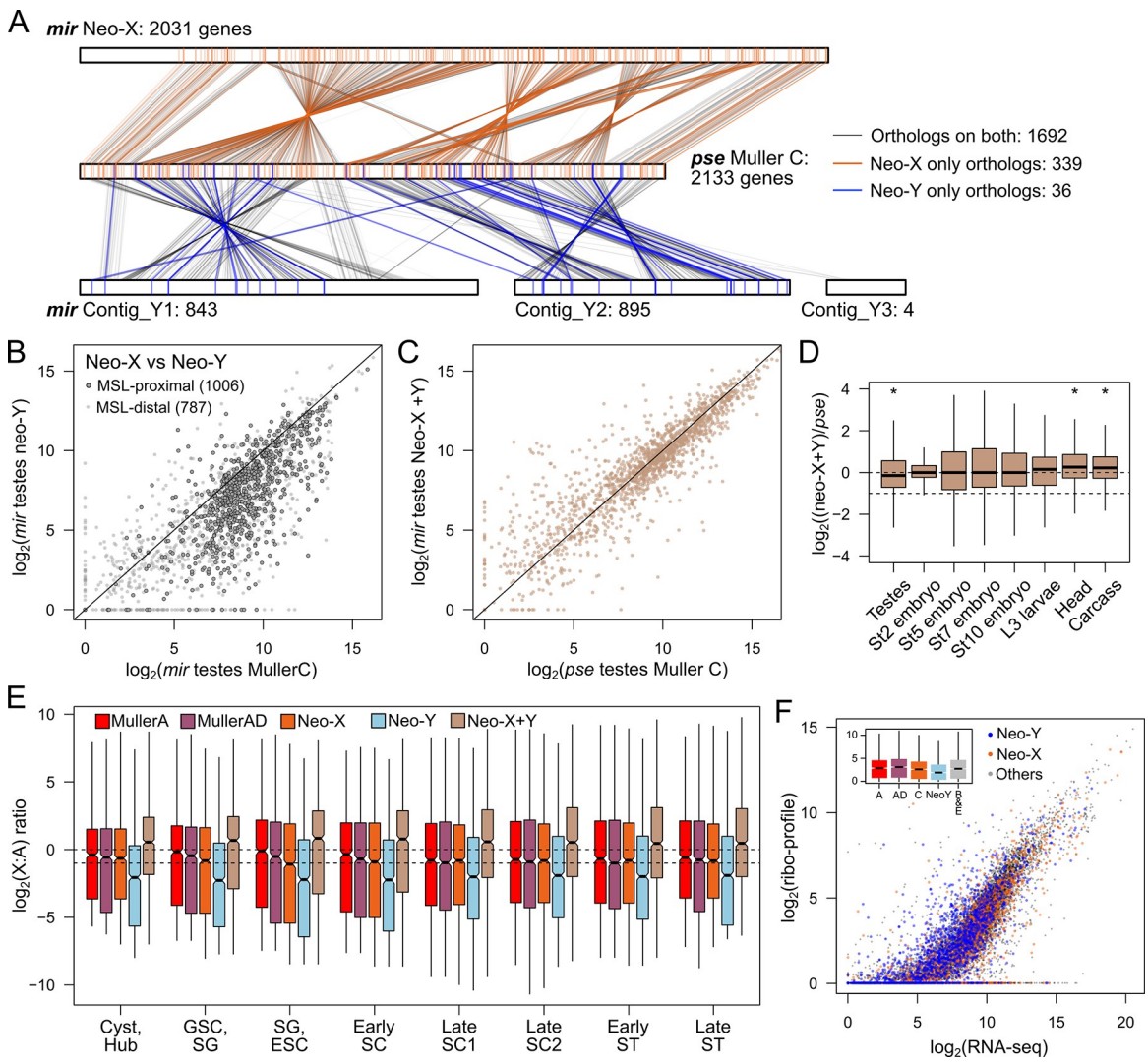

**Fig 5. Y-linked gametologs mitigates expression and protein dosage imbalance. (A)** Synteny and positions of Muller C genes before (*D. pseudoobscura*) and after (*D. miranda*) the formation of neo-sex chromosomes. Muller C from *D. pseudoobscura* (middle) is sandwiched by the neo-X (top) and neo-Y (bottom) in *D. miranda*. Genes where both gametologs (*n* = 1,692) are identified are connected with gray lines. Genes where only neo-X (*n* = 843) or neo-Y (*n* = 36) gametologs are present are connected by orange or blue lines, respectively. Number of genes on each chromosome/contig is noted next to the chromosome labels. **(B)** Testes expression correlation between neo-X and neo-Y gametologs. Diagonal line demarcates the identity line. **(C)** Testes expression correlation between *D. pseudoobscura* genes on Muller C and the combined expression of neo-X and neo-Y gametologs in *D. miranda*. **(D)** Log2(fold-difference) between the sum of *D. miranda* neo-X and neo-Y gametologs and *D. pseudoobscrua* Muller C-linked genes across different tissues and developmental stages. * = *p* < 0.000001, Wilcoxon's rank sum test. **(E)** Sum of the neo-X and neo-Y transcript count across testes cell types in the scRNA-seq data. **(F)** Correlation between the TPM from the RNA-seq data and ribosome profiling data from male larvae. The neo-X and neo-Y linked gametologs are plotted in orange and blue, respectively. Inset boxplot displays the distribution of the ribosome profiling reads across the chromosomes. The data underlying this figure can be found in S1 Data. scRNA-seq, single-cell RNA-seq.

their neo-X-linked and autosomal (Muller C) counterparts (**Figs 5B and S9**), neo-Y-only genes are expressed significantly higher than other Y-linked genes in the testes (**Fig 6A**; Wilcoxon's rank sum test, *p* < 0.005). Interestingly, these genes are similarly highly expressed in *D. pseudoobscura* testes (where they are autosomal) and lowly expressed in females (**Fig 6B**), suggesting an ancestral function in testis. In addition to the neo-Y-only genes, we further identified 115 genes where the neo-Y gametolog is significantly higher expressed than the neo-X

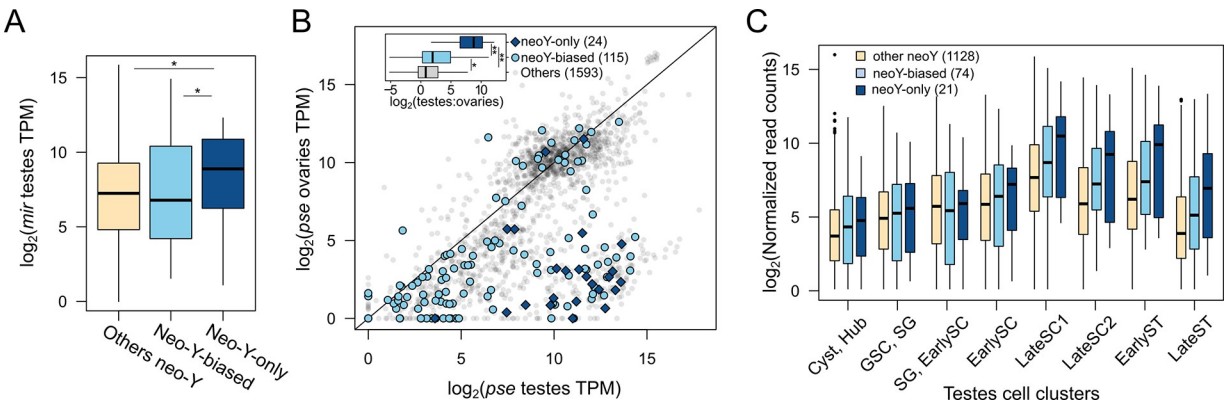

**Fig 6. Y-biased and exclusive genes are enriched for testes-biased genes resolving sexual antagonism. (A)** Testes expression of neo-Y genes that show expression bias (neo-Y-biased) or have no neo-X gametologs (neo-Y-only). * = $p < 0.0001$. **(B)** Expression correlation of Muller C-linked genes between testes and ovaries in *D. pseudoobscura*. Genes where the neo-Y gametolog is more highly expressed than the neo-X gametolog in *D. miranda* testes (neo-Y-biased) or has only Y-linked gametologs (neo-Y-only) are labeled by light blue circles and dark blue diamonds, respectively. Inlet displays boxplots of the log2(fold-difference) between *D. pseudoobscura* testes and ovaries for the different gene categories. * = $p < 0.001$, ** = $p < 0.00001$, Wilcoxon's Rank Sum Test. **(C)** Expression of Y-linked genes across testes and spermatogenesis cell types showing elevated and lasting postmeiotic expression. The data underlying this figure can be found in S1 Data.

gametolog (neo-Y-biased). Similarly, we find that these neo-Y-biased genes also show significant testes-biased expression in *D. pseudoobscura* where they are still autosomal, although not to the same extent as the neo-Y-only genes (**Fig 6B**). These results reveal that the X is becoming demasculinized through preferential loss of testes genes that become neo-Y-specific.

To specifically understand the germline regulation of these neo-Y-only and neo-Y-biased genes, we looked at their expression across the different testes cell clusters from the single-cell data (**Fig 6C**). We find that compared to other Y-linked genes, neo-Y-only genes consistently show the highest expression across all cell types. Strikingly, their expression disproportionally increases through meiosis, peaking at late spermatocyte and remains high upon meiotic exit and sperm individualization. Therefore, late and postmeiosis appear to be the primary stages in which these neo-Y genes are most active.

After the formation of neo-sex chromosomes, the neo-Y is transmitted exclusively through males and therefore expected to become masculinized. While genes with important male function may be preferentially retained on the Y, it is unclear why they are lost or down-regulated on the X. Genes with male-limited function could be lost on the X by random drift. Alternatively, these genes could represent sexually antagonistic alleles that are male beneficial and female detrimental and the neo-X linked copies are lost to eliminate deleterious effects in females. The formation of the neo-Y chromosome provides an opportune environment to resolve this conflict by allowing such genes to become male exclusive, alleviating deleterious effects in females.

Previous studies have revealed an excess of genes with testis-expression migrating off the X chromosome in *Drosophila* [72], and MSCI was invoked to explain this exodus. To determine whether a similar migration of X-linked testes genes to autosomes occurred on the neo-X of *D. miranda*, we identified orthologs between *D. miranda* and its sister species *D. pseudoobscura* where Muller C remains autosomal. We do not find an excess of Muller C genes to have moved to autosomes, albeit the number of gene movements is small (**Fig 5A**). Instead, we find that Muller A appears to be the most frequent destination for genes previously on the Muller C. Although small number size precludes meaningful statistical comparison ($p = 0.4018$, Fisher's exact test), prima facie, this is inconsistent with the need to escape X inactivation. The unique opportunity to resolve sexual antagonism provided by the neo-Y offers an alternative

path for genes with testes function to escape the X: After an autosome becomes sex-linked by forming neo-sex chromosomes, male-beneficial female-deleterious genes are lost on the X to become Y-exclusive. However, this Y-linkage is only a temporary solution as gene erosion and regulatory interference and epigenetic conflict from transposable elements will eventually drive these genes to either migrate off the Y or be lost entirely.

## Co-amplified neo-sex genes are predominantly expressed during meiotic stages of spermatogenesis

While Y evolution is characterized by massive gene loss and degeneration of the Y chromosome, we recently found that a subset of genes became highly amplified on both the neo-X and neo-Y chromosome of *D. miranda* after they became sex-linked [73]. Genes that co-amplified on the neo-sex chromosomes were highly enriched for functions associated with chromosome segregation, chromatin organization, and RNAi, suggesting that their amplification was driven by X versus Y antagonism for increased transmission. Consistent with the hypothesis that co-amplified genes functioning to interfere with mendelian segregation, we find that expression of both co-amplified X and Y genes drastically increases as they enter meiosis (peaking in late SC 1; **Fig 7A and 7B**). Co-amplified genes continue to be expressed postmeiosis when sperm chromatin is being remodeled, another process that has been found to be repeatedly targeted by meiotic drivers [25,74–76].

Expression patterns at co-amplified X/Y genes fall into 4 distinct clusters. Some genes are highly expressed from the neo-Y throughout spermatogenesis and peak during meiosis (Cluster 1); they typically have very high copy number on the Y (up to 39 copies), and only few copies on the neo-X. These genes may have amplified on the neo-Y because of important testis-specific functions they perform during meiosis. The second cluster consists of genes that are highly expressed from both the neo-X and neo-Y chromosome. Their expression peaks during

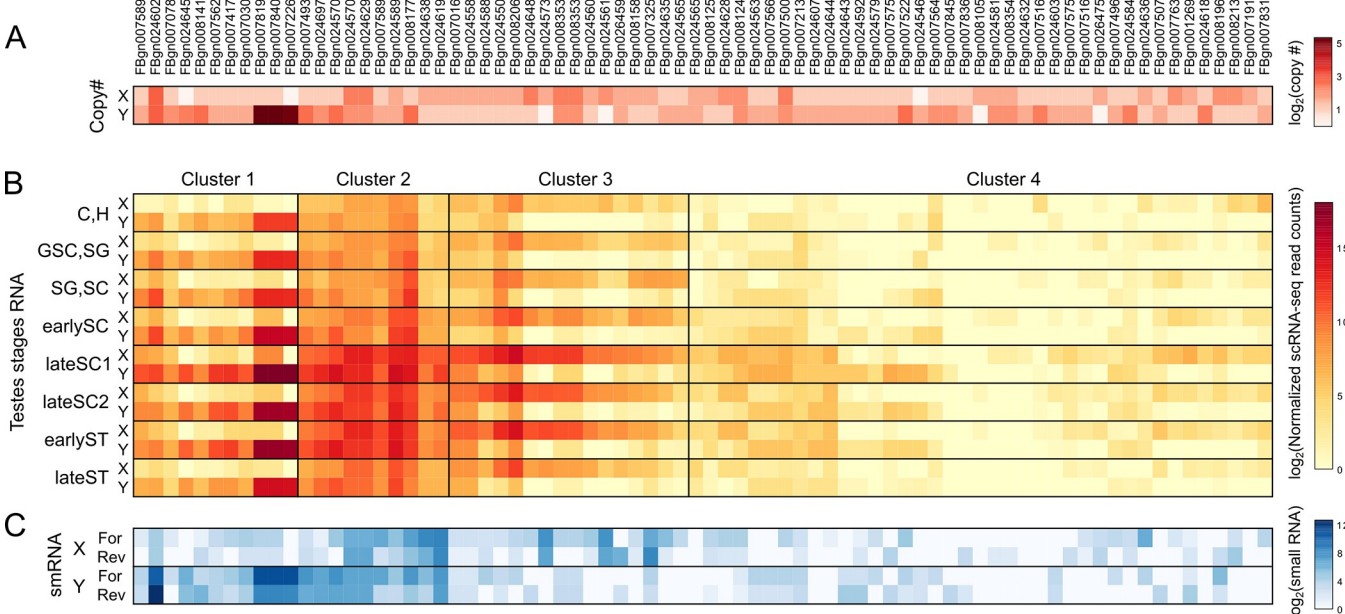

**Fig 7. Expression of ampliconic genes on the neo-sex chromosome across germline progression.** For each ampliconic gene family (columns), the copy number (**A**), expression across testes stages (**B**), and smRNA abundance (**C**) are displayed in heat maps in log scale. For scRNA-seq expression and smRNA, expression level is summed across X- or Y-linked copies. smRNA abundance is summed across all fragment sizes. For abundance across the entire fragment size range, see S12 Fig. The data underlying this figure can be found in S1 Data. scRNA-seq, single-cell RNA-seq; smRNA, small RNA.

and after meiosis (in late spermatocytes and early spermatids), and their copy numbers are more similar on the neo-X and neo-Y. We speculate that these genes are most likely to be involved in an ongoing, gene dosage-dependent conflict over segregation, where both X and Y copies are expressed during or after meiosis in a battle over transmission to the next generation. The third cluster consists of co-amplified genes predominately expressed from the neo-X copies; this could be "winning" X-linked drivers pushing the population sex ratio towards females, as observed in wild *D. miranda* populations [73]. The last cluster contains co-amplified genes that show little expression from either chromosome. It is possible that these genes are now defunct relics of previous meiotic drive systems that are now effectively silenced.

Since the siRNAs pathway has been shown to suppress sex-linked meiotic drivers [77–79], we analyzed small RNA (smRNA) profiles from bulk testes [73]. Interestingly, X-linked and Y-linked multicopy genes with high copy number and expression produce an abundance of both sense and antisense smRNAs (**Figs 7** and **S11**). Further subdividing the smRNA reads based on their size, we find that the smRNAs are predominantly distributed around 21 bp indicative of siRNAs (and not simply RNA degradation products; **S12 Fig**). These results are consistent with the scenario that these sex-linked multicopy genes are reciprocally down-regulating each other by producing siRNAs to bias their transmission into the next generation.

## Discussion

### Dosage compensation shutdown through meiosis explains meiotic and germline down-regulation in flies

The status of meiotic sex chromosome regulation in *Drosophila* has been debated for decades. The first evidence for MSCI in flies, albeit indirect, was based on observations of reciprocal translocations between the X and autosome. Autosome-to-X translocations cause male sterility but not X-to-autosome translocations, suggesting that germline-essential genes translocated onto the X are down-regulated [80]. In conjunction with claims of increased X-condensation in early spermatocytes, this led to the suggestion of MSCI in flies [80], but the cytological data were subsequently refuted [81]. More recent, direct evidence of MSCI includes reduced expression of transgenes inserted onto the X versus autosomes [28,34], and reduced expression of X-linked genes in carefully dissected testes tissues as assayed by qPCR and microarrays [29,30]. In conjunction with the observation that genes with testes-biased expression are underrepresented on the X in flies, it has been proposed that MSCI drives the exodus of testes-biased genes off the X chromosome onto autosomes [29,33]. However, others have argued that the X is not globally inactivated [31], or that the reduced expression is not exclusive to meiotic stages but occurs even in premeiotic cells [30,34]. Discrepancies have been attributed to limitations in methodologies including limited number of genes quantified and tissue contamination in manual dissections, issues that scRNA-seq holds the promise of resolving. When analyzing the meiotic cell types in adult *Drosophila* testes scRNA-seq, Witt and colleagues were unable to establish the presence of MSCI, reporting spermatocyte expression levels that are "expected from incomplete or absent dosage compensation" [35]. However, Mahadevaraju and colleagues reported the presence of MSCI after analyzing scRNA-seq generated from developing testis discs in male larva [36], although the reported down-regulation does not exceed half of autosomal expression (and thus is also consistent with absence of DC in testes). Therefore, even with scRNA-seq relieving concerns of cell-type contamination the status of MSCI in *Drosophila* remains debated.

We took advantage of *D. miranda's* unique sex chromosome architecture to evaluate the presence of MSCI in conjunction with scRNA-seq. *D. miranda's* 3 X chromosome arms of different ages allowed us to determine whether and how the X chromosome is down-regulated

during spermatogenesis, and how it evolves. Inconsistent with the X being inactivated during meiosis, expression of X-linked genes increases through meiosis, although to a lesser extent compared to the autosomes (**Fig 1F**). Furthermore, overall expression of the X's never drops below half that of the autosomes. Contrasting expression trajectories of the different X chromosomes is particularly diagnostic as to what process influences X expression during spermatogenesis, since DC shutdown and MSCI make different predictions for X arms with different ages (**Fig 2A**). If the X is silenced by MSCI, the neo-X is expected to be the least down-regulated due to its most recent autosomal heritage, while under DC shutoff expression of all 3 X's is expected to simply equalize (but not drop below half of autosomal expression levels). Prior to meiosis, the neo-X consistently has the lowest expression of all 3 X's, indicating incomplete dosage compensation. However, we find that the neo-X loses expression the quickest and through meiosis, expression of older X's simply drops to comparable levels. Expression levels of all 3 X arms equalize after meiotic exit at roughly 58.5% of autosomal expression (**Fig 2B**). These results collectively argue that the X's are losing dosage compensation during spermatogenesis, resulting in relatively lower expression compared to autosomes. One critical benefit of examining the neo-X in *D. miranda* is the possibility of inferring the ancestral expression prior to sex linkage by analyzing autosomal orthologs in *D. pseudoobscura* and neo-Y linked gametologs. Both comparisons revealed that low expression of a subset of neo-X genes most likely reflects ancestrally low expression, instead of acquired repression after X-linkage (**Fig 3B and 3C**).

This model of relative X down-regulation through DC loss reconciles discrepancies in several studies. It is fully consistent with both the scRNA-seq findings in Witt and colleagues [35] and Mahadevaraju and colleagues [36]. In the latter study, cytological data showing reduced active RNA-Polymerase-II (Pol-II) staining in meiotic nuclei was argued as critical support for MSCI-driven down-regulation. In light of our results, we suspect that reduced Pol-II activity reflects the attenuation or removal of DC. MSL-associated hyper-transcription in somatic cells is achieved via elevated levels of Pol-II recruitment to promoters [82] and enhanced elongation [83,84]. The cytological results in Mahadevaraju and colleagues [36] show that active Pol-II signals on the X, estimated as the ratio between active and total Pol-II signals, appear to be around half of the ratio for autosomes, suggesting that half of the Pol-II occupying the X are no longer active. While this was interpreted as MSCI, it could also represent reduction of Pol-II activity on the hyper-transcribed X without clearing Pol-II occupancy, and therefore is fully consistent with DC shutdown. Moreover, their volumetric estimates suggest that the X is less compacted and occupies more nuclear space than autosomes; this is inconsistent with the compaction of the X as expected of and observed in animals with MSCI. Interestingly, a recent study by Anderson and colleagues [85] profiled active Pol-II in spermatocytes of meiotic arrest mutants and reported that genome-wide active Pol-II profiles across the X and autosomes are fully consistent with a lack of dosage compensation by this stage of spermatogenesis [85]. They also found no evidence of silencing chromatin enrichment on the X in spermatocytes and similarly concluded the lack of DC as the cause of X down-regulation. Our results are also in partial agreement with the reports in Meiklejohn and colleagues and Landeen and colleagues that down-regulation of the X occurs across the testes and is not specific to meiosis [30,34], as the X:A expression ratios in somatic (Cyst and Hub) and early germline clusters (GSC and SG) are all <1 and thus lower than expected under full dosage compensation (**Fig 2B**).

## Germline dosage compensation is less robust on the neo-X

Our analyses comparing X-linked genes proximal and distal to the MSL complex clearly shows DC up-regulating the former in many of the testis clusters until late meiotic cell types (late

spermatocytes) (**Fig 1F**). While this is consistent with previous reports in *D. melanogaster* testes scRNA-seq dataset [35,36], the germline is thought to lack MSL-dependent dosage compensation in flies. This is primarily based on the cytological observation that many of the MSL components are absent in the germline, the male germline appears to lack X-chromosome-specific H4K16ac [37] and MSL-1 and MSL-2 are dispensable for spermatogenesis in pole cell transplants [86]. However, we find that only *roX1* and *mof* are absent or lowly expressed in the germline tissues of *D. miranda*. *Rox1* and *rox2* have essential but redundant function in dosage compensation and *mof*, the histone acetyltransferase, appears to be lowly expressed even in other scRNA-seq datasets for somatic tissues that are expected to be dosage compensated [87,88], so low expression of *mof* may not necessarily reflect absence of activity. Thus, our data are consistent with MSL-dependent DC in the testis of *D. miranda*, but alternative means of achieving X up-regulation in the GSC and spermatocytes without hyper-acetylation are possible. Three-strand DNA-RNA-hybrid structure (R-loops) were shown to be associated with hyper-transcription on the X in the absence of H4K16Ac [89]. Further, a recent comparison of the nuclear topology of germline stages reported that in spermatogonia and spermatocytes, active chromatin regions on the X have more intra-chromosomal contacts than the autosomes, potentially suggesting that active regions on the X are gathering at transcription-rich nuclear territories allowing for higher expression [38]. These studies both suggest alternative mechanisms of up-regulating the X in the absence of the full MSL complex.

Curiously, the (perhaps, non-canonical) dosage compensation in the germline appears to differentially up-regulate the X arms. The neo-X shows incomplete dosage compensation across all tissues, even somatic ones (**Fig 4B**). However, incomplete dosage compensation becomes much more pronounced in the early germline (**Fig 2B** and 2**C**), where the extent of dosage compensation tracks the age of the X's. This suggests that germline DC is either less robust or being lost more quickly on the younger neo-X.

While dosage compensation of the neo-X appears most incomplete in testis, we also found that expression of the neo-Y alleles can increase overall expression from the neo-sex chromosomes (**Fig 5D**). By examining the expression of gametologs on the neo-sex chromosomes in the scRNA-seq and comparing their expression to the ancestral autosomal expression in bulk RNA-seq data, we found that the neo-Y chromosome can mitigate suboptimal dosage compensation on the neo-X, regardless of the tissue type. Importantly, since neo-Y alleles are more highly expressed in testis compared to most autosomal tissues (**S13 Fig**), this means that overall expression from the neo-sex chromosomes in testis resembles that of the older X chromosomes more closely—i.e., only a moderate deficiency of Muller C linked expression relative to autosomes in testis, similar to that of the older X chromosomes.

## Gene content and gene expression evolution resolves sexual and meiotic conflict of sex chromosomes

Gene content of sex chromosomes is non-random, and gene expression evolution and gene movement can contribute to sex chromosome-specific expression patterns. We find that the neo-Y is becoming masculinized. Ancestrally testes-biased genes (inferred from expression patterns in *D. pseudoobscura*) have become either neo-Y biased in expression or neo-Y exclusive, indicating either down-regulation or complete loss of the neo-X gametologs, respectively. Demasculinization of the X chromosome occurs, in part, due to the fact that the X chromosome spends twice as much time in females than males, and previous studies in *Drosophila* have revealed an excess of gene movement of testis-expressed genes off the X chromosome onto autosomes [6,42]. Here, we show that the neo-Y presents an intermediary for this movement as male-biased genes first transition to neo-Y biased or specific expression, thereby

alleviating the potential for these genes to be female-detrimental and resolving sexual conflict. However, as degeneration continues, these genes will inevitably be lost, further driving their movement onto autosomes. Interestingly, the formation of the neo-X in *D. miranda* is not associated with an excess of gene movements from Muller C to other autosomes (**Figs 3A** and **4A**); in fact, gene movement is small and Muller A appears to be the most frequent destination. Thus, autosomal migration likely occurs only after further degeneration of the neo-Y which provides a temporary but unstable solution to sexual conflict.

Sex chromosomes are also vulnerable to be involved in conflicts during meiosis. Meiotic drivers that try to cheat fair mendelian segregation are prone to originate on sex chromosomes. Sex chromosome drive will skew the population sex ratio and select for suppressors on the other sex chromosome that are resistant to the distorter. If the distorter and suppressor are dosage sensitive, they would undergo iterated cycles of expansion, resulting in rapid co-amplification of driver and suppressor on the X and Y chromosome. We recently showed that meiosis and RNAi-related genes co-amplified on both the neo-X and neo-Y chromosome of *D. miranda*. Consistent with co-amplification of X/Y genes being driven to interfere with mendelian segregation, we show that these genes are predominantly expressed during meiotic stages of spermatogenesis and postmeiotic stages, the cell types in which meiotic drivers are most likely to operate. Thus, sexual and meiotic conflicts may drive unique gene content and expression evolution on sex chromosomes.

## Materials and methods

### Testes disruption and single-cell preparation

For each of the 2 replicates, 5 pairs of testes were dissected from *D. miranda* MSH22 in drops of cold PBS. Testes were transferred to a low retention Eppendorf tube containing a drop of lysis buffer (Trypsin L + 2 mg/ml collagenase) followed by 30 min of room temperature incubation with mild agitation. Then, the samples were first passed through a 50 μm nylon mesh filter (Genesee Cat #: 57–106) with centrifugation at 1,200 rpm for 7 min, and then twice through a 30 μm nylon mesh (Genesee Cat #: 57–105). The resulting cell pellet was washed with 200 μl of cold HBSS and pelleted again by centrifuging at 1,200 rpm for 7 min. After removing the supernatant, the cells were resuspended in 20 μl of HBSS. Approximately 5 μl of the cell suspension were used for cell counting, and 5 μl were transferred to a slide for imaging with a Zeiss upright light microscope. Approximately 10,000 cells were submitted for library preparation with the 10X Chromium platform at the Berkeley Vincent J. Coates Genomics Sequencing Lab followed by 100-bp pair-end sequencing on Illumina NextSeq 2000.

### Single-cell RNA-seq data processing

We used salmon (v1.5.2) [90] package to align the raw reads. We generated the index using a fasta file containing both *D. miranda* genic and transposable element transcripts. We then mapped the reads using the alevin program in salmon forcing 10,000 cells with—forceCells. Resulting files were then imported into R (v4.2.1) in Rstudio. We used Seurat (v4.0.5) [91] to process the read counts, cluster the UMIs, assign cell identities, generate UMAP projections, and aggregate cells within cell types. While there were initially 10,990 UMIs, after clustering and cell type assignment using marker genes [35,36,62,63,92], we were unable to assign cell identity for 4 clusters that have low read counts (**S1 Fig**). Cells (5101) in these clusters were removed, resulting in a final cell count of 5,889. To remove non-testes-related genes, we removed genes with fewer than 5 total reads across any cell types. All subsequent analyses were done in Rstudio.

### Cluster-specific expression and X:A ratio

We used the function AggregateExpression() in Seurat to sum counts across cells of the same cluster to acquire cluster-specific expression of every gene. To calculate the X:A ratio, each X-linked gene is divided by the median expression of genes on Muller B and E, generating expression distributions that are autosome-normalized. The cell-type and chromosome-specific medians were used to infer the DC status chromosome-wide or for specific gene sets.

### MSL3-binding site and dosage compensation identification

The MSL-3 ChIP-seq is from ref [2] which were generated by pulling down a GFP-tagged version of MSL-3 in D. miranda. We used bwa mem on default settings [93] (v 0.7.15) to align the chip and input reads. We then used macs2 [94] (v2.2.6) to call the enrichment peaks. We then used bedtools [95] (v2.26.0) closest -d to identified the gene closest to MSL3 peaks and vice versa and their distance from each other, allowing us the categorize genes according to their proximity to MSL peaks.

### Identifying orthologs and gametologs using reciprocal best hits

We downloaded the *D. pseudoobscura* (r3.2) CDS sequence from Flybase. For *D. miranda*, we generated 2 sets of CDS files, one without neo-X genes and one without neo-Y genes. For each gene, the longest CDS is selected if there are alternative splice forms. We then used blastn (v2.6.0+) -task blastn to reciprocally blast the *D. pseudoobscura* CDS sequences with either sets of *D. miranda* CDS sequences. Best blast hit was defined as either having the highest e-value or longest blast alignment. Genes were identified as orthologs if they are reciprocal best hits of each other. The neo-Y gametologs are the Y-linked reciprocal best hits in the *D. miranda* CDS without the neo-X genes. The neo-X gametologs are the Muller C-linked reciprocal best hits in the *D. miranda* CDS without the neo-Y genes. Genes were further blasted to the *D. lowei* genome [96] to determine the ancestral chromosome location. Ampliconic genes were removed for orthology calls.

### Ribo-seq library preparation

We collected male third instar larvae and extracted total RNA using the RNAeasy Mini kit (Qiagen Catalog No. 74004). We digested the total RNA by micrococcal nuclease (Roche) with 3U/μg of total RNA. To collect the monosomes, we prepared 10% to 50% sucrose gradients in polysome gradient buffer (250 mM NaCl, 15 mM $MgCl_2$, 20 U/ml Superase ln, 20 μg/ml Emetine) with a GradientMaster (Biocomp Instruments). Monosome fractions were then collected after resolving and fractionating the gradient. We then extracted the RNA from the monosomes with a standard phenol/chloroform protocol and dephosphorylated the RNA by T4 polynucleotide kinase (New England Biolabs). We specifically excised RNAs spanning 28 to 34 nt from the gel, and then performed 2 rounds of subtractive hybridization to remove ribosome RNAs as described. Small RNA libraries were then prepared following the standard Illumina protocol with TruSeq small RNA kit (Illumina Catalog No. RS-200). The ribo-seq libraries were sequenced at the Vincent J. Coates Genomics Sequencing Laboratory at University of California, Berkeley at HiSeq 2000 50 bp SE.

### RNA-seq and ribo-seq data and processing

Embryonic and tissue-specific RNA-seq were from refs [97,98]. Reads were aligned using STAR [99] (v2.7.10a) to their respective species genomes. The *D. miranda* genome is from ref [57] and the *D. pseudoobscura* genome (r3.2) is from Flybase. We then filtered the mapped

reads for either the primary alignments or the unique alignments using samtools view–F 260 or–q 255, respectively. We then used featureCounts [100] (v2.0.3) in the Subread package to count reads over CDSs with featureCounts -p -M -t CDS. Because the X and Y can have large differential contribution to male and female transcriptomes, for normalization we used a modified version of Transcript Per Million with number of reads mapped to autosomal genes as the numerator: ((No. of reads mapped to a gene / CDS length*10e-6)) / (Sum of (No. of reads mapped to autosomal gene / CDS length) * 1000) * 10e-6).

SmRNA data processing: Testes smRNA were from [73]. Reads were trimmed with trim_-galore (http://www.bioinformatics.babraham.ac.uk/projects/trim_galore/). They were mapped using bowtie allowing only 1 mismatch. Mapped reads were then fractionated into different fragment sizes using awk. We then used featureCounts–p–t CDS–s to count reads mapping to CDSs in a strand aware manner.

## Dryad DOI

doi.org/10.6078/D1DH7T [101].

## Supporting information

**S1 Table. scRNA-seq summary.**
(XLSX)

**S2 Table. Gene filtering approaches.**
(XLSX)

**S3 Table. Number of genes at different distances to MSL peaks.**
(XLSX)

**S1 Fig.** (A) UMAP projection of all UMI clusters. (B) Expression of maker genes used to identify cell types across clusters. The group of genes expressed in the spermatocyte necessary for meiosis (collectively known as the meiotic arrest genes) are labeled in red. The data underlying this figure can be found in S1 Data.
(PDF)

**S2 Fig. Log scale distribution of expression fold differences between different cell stages for different chromosomes.** $^* = p < 0.000001$, Wilcoxon's rank sum test. The data underlying this figure can be found in S1 Data.
(PDF)

**S3 Fig. Subclustering of spermatocyte stages.** (A) By using a more sensitive clustering parameter, additional clusters were identified for the spermatocytes which may correspond to different sperm types. (B) X:A ratio of the sex chromosomes in the subclusters. The data underlying this figure can be found in S1 Data.
(PDF)

**S4 Fig. X:A ratios of different gene sets.** (A) Testes-expressed genes across meiotic stages: genes in the scRNA-seq data were selected based on bulk testes RNA-seq. Testes-expressed genes are classified as those expressed in the upper 50% of the expression (TPM) distribution. (B) All genes were used if they show more than 10 reads in the scRNA-seq experiments. The data underlying this figure can be found in S1 Data.
(PDF)

**S5 Fig.** (A) X:A ratio depending on distance from MSL peaks. (B) Normalized read counts of the Xs depending on distance from MSL in the early testes stages. The data underlying this

figure can be found in S1 Data.
(PDF)

**S6 Fig. Distinct transcriptional landscape across meiotic progression for genes distal and proximal to MSL.** Genes are divided based on their distance from MSL peaks (columns 1–4). Across meiotic stages (rows), log2 scale gene expression is normalized by the median autosomal counts, and their distributions are depicted as kernel densities. Column 5 shows the distribution of autosomal genes found on Muller B and E. Vertical lines within plots demarcate X:A ratios of 1. The data underlying this figure can be found in S1 Data.
(PDF)

**S7 Fig. Testes expression difference between neo-X (*D. miranda*) and autosomal (*D. pseudoobscura*) orthologs.** The data underlying this figure can be found in S1 Data.
(PDF)

**S8 Fig. Expression difference between Muller C orthologs in female tissues.** The data underlying this figure can be found in S1 Data.
(PDF)

**S9 Fig. Muller-C expression in *D. pseudoobscura* vs. neo-Y expression in *D. miranda*.** The data underlying this figure can be found in S1 Data.
(PDF)

**S10 Fig. Ribosome profiling and occupancy of *D. miranda* male larvae.** (A) Ribosome profiling reads in TPM for neo-Y genes with untruncated or truncated CDS. Neo-Y genes are considered truncated if the CDS is shorter than the neo-X CDS by 20%. (B) Ratio of ribosome profiling reads of sex chromosomes to autosomes. (C) Ribosome occupancy as measured by the ratio ribosome profiling reads (in TPM) over RNA-seq reads (in TPM) for the chromosomes. $* = p < 0.0001$. The data underlying this figure can be found in S1 Data.
(PDF)

**S11 Fig.** Contrast between multicopy genes on the neo-X and neo-Y for copy number (A), abundance of sense smRNA (B), and abundance of antisense smRNA (C). The data underlying this figure can be found in S1 Data.
(PDF)

**S12 Fig. smRNA abundance for X and Y linked multicopy genes broken down into fragment sizes corresponding to different types of small RNAs.** The data underlying this figure can be found in S1 Data.
(PDF)

**S13 Fig.** (A) Expression of neo-Y-linked genes across developmental stages and tissue types. $* = p < 2.2e-16$, Wilcoxon's rank sum test when compared to testes expression. (B) Expression of Y-specific genes across developmental stages and tissue types in *D. pseudoobscura* and *D. miranda*. The data underlying this figure can be found in S1 Data.
(PDF)

**S1 Data. Data and code for analysis reproduction.**
(ZIP)

## Acknowledgments

We thank the anonymous reviewers and the Academic Editor for their insightful feedback.

## Author Contributions

**Conceptualization:** Doris Bachtrog.

**Data curation:** Kevin H-C. Wei, Kamalakar Chatla.

**Formal analysis:** Kevin H-C. Wei.

**Funding acquisition:** Doris Bachtrog.

**Investigation:** Kamalakar Chatla.

**Methodology:** Kevin H-C. Wei, Kamalakar Chatla.

**Project administration:** Doris Bachtrog.

**Resources:** Doris Bachtrog.

**Supervision:** Doris Bachtrog.

**Validation:** Kevin H-C. Wei.

**Visualization:** Kevin H-C. Wei.

**Writing – original draft:** Kevin H-C. Wei, Doris Bachtrog.

**Writing – review & editing:** Kevin H-C. Wei, Doris Bachtrog.

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
