## [Editor Report · Decision Letter 0]

7 Jan 2023

Dear Dr Bachtrog, 

Thank you for submitting your manuscript entitled "Single cell RNA-seq in Drosophila testis reveals evolutionary trajectory of sex chromosome regulation" for consideration as a Research Article by PLOS Biology. Please accept my sincere apologies for the long delay in getting back to you owing to the recent Christmas holiday period. 

Your manuscript has now been evaluated by the PLOS Biology editorial staff, as well as by an academic editor with relevant expertise, and I am writing to let you know that we would like to send your submission out for external peer review.

Once your full submission is complete, your paper will undergo a series of checks in preparation for peer review. After your manuscript has passed the checks it will be sent out for review. To provide the metadata for your submission, please Login to Editorial Manager (https://www.editorialmanager.com/pbiology) within two working days, i.e. by Jan 09 2023 11:59PM.

Kind regards,

Richard

Richard Hodge, PhD

Associate Editor, PLOS Biology

rhodge@plos.org

PLOS

---

## [Decision Letter · Decision Letter 1]

14 Mar 2023

Dear Doris,

Thank you for your continued patience while your manuscript "Single cell RNA-seq in Drosophila testis reveals evolutionary trajectory of sex chromosome regulation" was peer-reviewed at PLOS Biology. Please accept my sincere apologies for the long delays that you have experienced during the peer review process. Your manuscript has now been evaluated by the PLOS Biology editors, an Academic Editor with relevant expertise, and by three independent reviewers. I have also provided some specific comments from the Academic Editor below the reviewer reports (labelled 'Comments from the Academic Editor').

In light of the reviews, which you will find at the end of this email, we would like to invite you to revise the work to thoroughly address the reviewers' reports.

As you can see, Reviewers #2 and #3 are positive about your study and are mostly satisfied that the conclusions regarding the importance of a loss of dosage compensation during X-linked gene expression regulation are supported by the data. On the other hand, Reviewer #1 raised several concerns with the overall strength of the evidence to support the dosage compensation model, including the lack of cytological assays to validate the expression of testes markers in D. miranda. After discussions with the Academic Editor, we will not make these experiments essential to consider a revised version, but we do encourage you to include this data if you have already performed the validations or to caveat and address any assumptions made in the manuscript text. As you will see in the ‘Comments from Academic Editor’ provided beneath the full reports, the Academic Editor has also provided a few additional suggestions that he/she feels are important to address in order to provide further contextualization for the work and clarify some potential limitations. 

Given the extent of revision needed, we cannot make a decision about publication until we have seen the revised manuscript and your response to the reviewers' comments. Your revised manuscript is likely to be sent for further evaluation by all or a subset of the reviewers.

**IMPORTANT - SUBMITTING YOUR REVISION**

*Re-submission Checklist*

*Published Peer Review*

*PLOS Data Policy*

*Blot and Gel Data Policy*

Sincerely,

Richard

Richard Hodge, PhD

Associate Editor, PLOS Biology

rhodge@plos.org

REVIEWS:

Reviewer #1: Drosophila miranda is a rich resource for understanding the evolution of X chromosomes, and it may shed light on this chromosome's epigenetic trajectory. However, in my opinion, this study is not appropriate for publication in PLoS Biology because the lack of cytological experiments may lead the authors to draw insufficient or incorrect conclusions. The field of MSCI in Drosophila has already been the subject of lengthy debate, largely due to similar issues that hampered scientific progress. The Sex Chromosome Meiotic Inactivation is recognized as an epigenetic phenomenon with cell detectable consequences, and cytology experiments helped to clear up previous misunderstandings.

Below I explain my evaluation in detail. 

Major points

1) Cytological experiments are necessary for detecting and classifying cell clusters in Drosophila miranda:

a) In comparison to the majority of the other species in the genre, the Drosophila pseudoobscura group has a distinct testis morphology. The organ's external format is a result of different spermatogenesis development in terms of shape, size, and process. For starters, instead of four mitotic divisions, the species have five and the final product is 128 spermatozoa for each initial cell. Second, two types of spermatozoa are produced. It means that this group is known to have sperm polymegaly, a type of sperm heteromorphism characterized by the production of two different sizes of spermatozoa within the same animal, with the only difference being the lengths of the head and tail. Furthermore, genes expressed in the testes of D. pseudoobscura have been found to be less similar to their D. melanogaster orthologs (PMID: 15632085). 

b) As a result, genetic markers with known expression in different testis cell phases of Drosophila melanogaster should not be used for definitive cluster classification. In the absence of a Drosophila miranda testis expression marker, authors should rely on genes that are only expressed in specific clusters for in situ hybridization confirmation of their predictions.

c) Clusterization and confirmation are not working properly with the author's current method, as shown in Figure 1 and Supplementary Figure 1. First, there are four unknown clusters with strikingly similar expression profiles. Second, the progression from late SC1 to ST exhibits the same problem. There is no evidence of polymegaly progression. As similar patterns of expression can lead to misclassification and thus further misinterpretation about timing and specific level and stage of X chromosome expression regulation, cytological characterization could shed light on these issues.

2) Cytological experiments are needed to differentiate between mRNA level and transcription

a) Because cells have a fixed rate of RNA degradation, mRNA expression measurements are a proxy for transcription. However, drosophila spermatogenesis is known for producing a large number of proteins that are required later in the post-meiotic stages and are translated from mRNAs produced during meiosis and stored in the cytoplasm (Olivieri and Olivieri 1965, Schäfer et al 1995). That is why the proportion of X/A should not be used alone to distinguish between MSCI and a lack of dosage compensation. mRNAs transcribed in previous stages are stored inside cells during later stages of development, which may prevent mRNA sequence methods from detecting transcriptional shut off. Cytological experiments have already been performed in Mahadevaraju S. et al, 2021, demonstrating the cytological absence of active RNA polymerase on the X chromosome territory in D. melanogaster spermatocytes and are a promising resource to solve this problem D. miranda. 

b) Also, it is not possible to use Inference of transcriptional kinetics parameters as done by the authors. In the model of transcription used, it is assumed that a gene switches to the on state and off state at exponentially distributed times with rates, and, regardless of gene state, the RNA is degraded with fixed rate which is not true for drosophila spermatogenesis. 

c) The authors have mentioned in the discussion:

 "Discrepancies have been attributed to limitations in methodologies including limited number of genes quantified and tissue contamination in manual dissections, and single cell RNA-seq should be able to resolve the debate. Analyzing the meiotic cell types in adult Drosophila testes scRNA-seq, Witt et al found no evidence of MSCI [35]. However Mahadevaraju et al reported the presence of MSCI after analyzing scRNA-seq generated from developing testis discs in male larva [36]. Therefore, even with scRNA-seq the status of MSCI in Drosophila remains unclear."

 First, it is a misinterpretation to think that Witt et al. and Mahadevaraju et al. came to a different conclusion. The first group found Dosage compensation in mitotic cells and the second group found the same results and also MSCI in meiotic cells. The former results were based on cytological evidence of lack of active RNA polymerase on the X in spermatocytes. Second, if, even scRNA-seq was not able to solve the MSCI debate in Drosophila (because of the reasons explained in 2A), why do the authors think that the same technique will shed a light on MSCI status in D. miranda? Third, it is surprising that an important work such as the one done with cytological experiments is only mentioned for the first time in this study's discussion.

3) MSCI vs. lack of Dosage compensation

a) If the authors had acknowledged previous cytological studies, they could have avoided making assumptions about elaborating their hypothesis. According to the authors, MSCI would predict a positive correlation between age and MSCI level in terms of X chromosome evolutionary history (Figure 1A). Based on cytological data, Mahadevaraju S. et al, 2021 proposed a territory-regulation model for X inactivation that could result in the same level of down regulation of all three different aged chromosomes if they were located in the same territory.

b) There is a lot of information that can be extracted from Figure 1. 

b.1) Data normalization is critical in rna-seq experiments. As a result, using different groups of genes would aid in determining how robust the result interpretation is. For example, counting genes that are normally not or are only weakly expressed in the testis will reduce the average and median expression of each chromosome. As a result, you will most likely underestimate the denominator (autosomal expression), and your X/A ratio will be overestimated. 

b.2) The use of MSL-distal and proximal regions is an intriguing approach, but the data can be explained by factors other than a lack of dosage compensation for X down regulation in meiosis. Male-biased genes, for example, are known to be significantly further away from a HAS than female- and unbiased genes. Male biased genes are typically enriched in testis-biased genes derived from meiotic expression, which is where the transcription burst occurs. As a result, the genes most affected by inactivation may be those MSL-distal simply because they are expressed more in the testis, as shown in Figure 2E, rather than due to a lack of dosage compensation. It is important to note that the MSL distal and proximal regions were created using Chip-seq data that were made with other tissue than testis. 

b.3) Following the authors' logic that Muller C has incomplete dosage compensation in general, and given that MSL-distal regions have the lowest dosage compensation hyper transcription effects, genes in Muller C - MSL-distal could be the best control to test MSCI vs. lack of dosage compensation. Figure 2E shows that in early SC, the median expression of genes in Muller C - MSL-distal is about 10% of autosomal median expression. This appears to be due to a stronger downregulation, such as MSCI, rather than a simple lack of dosage compensation.

b.4) Furthermore, if meiotic cells are losing dosage compensation, why does the level of expression of MSL-distal genes increase from late SC on? Perhaps it is transitioning from X inactivation to a non-dose compensated level. It is important to note that in a model where X inactivation occurs, dosage compensation should not be in place. 

4)The sessions "Neo-Y gametologs mitigate neo-X dosage imbalance" and "Neo-Y as an environment to resolve sexual conflict" are the most interesting of the study, but there are some issues.

a) The findings that NeoX+NeoY restores equality with autosomal ratio is interesting. However, for all tissues, these values are greater than one. Indeed, Y linked genes in Drosophila melanogaster (as opposed to other species) have a later peak of expression (meiosis and post-meiosis) in male meiosis due to a chromatin decondensation loop. D. miranda isn't particularly interesting because of its Neo-X and Neo-Y chromosomes. D. pseudoobscura species have a different Y chromosome than D. melanogaster (older than neo X, the YD). Furthermore, ancestral Y single copy genes were transposed to a region that is now fused to the muller F gene (the dot chromosome). They should look at expression of genes in YD and ancestral Y-linked genes to test the hypothesis that neo-Y linked expression may mitigate dosage imbalance from incomplete dosage compensation or loss in the testes.

b)

"Previous studies have revealed an excess of genes with testis-expression migrating off the X chromosome in Drosophila [72], and MSCI was invoked to explain this exodus. To determine whether a similar migration of X-linked testes genes to autosomes occurred on the neo-X of D. miranda, we identified or

---

## [Editor Report · Decision Letter 2]

15 Jan 2024

Dear Dr Bachtrog,

Thank you for your patience while we considered your revised manuscript "Single cell RNA-seq in Drosophila testis reveals the evolution and trajectory of germline sex chromosome regulation" for publication as a Research Article at PLOS Biology. This revised version of your manuscript has been evaluated by the PLOS Biology editors and the Academic Editor.

Based on our Academic Editor's assessment of your revision, we are likely to accept this manuscript for publication, provided you satisfactorily address the remaining points raised by the Academic Editor, and the following data and other policy-related requests.

IMPORTANT - please attend to the following:

a) Please include the species name in the Title, i.e. "Single cell RNA-seq in Drosophila miranda testis reveals the evolution and trajectory of germline sex chromosome regulation"

b) Please address the remaining requests from the Academic Editor (see the foot of this email).

c) Please address my Data Policy requests below; specifically, we need you to supply the numerical values underlying Figs 1DEF, 2ABCDE, 3ABC, 4ABCDEF, 5ABCDEF, 6ABC, 7ABC, S1AB, S2, S3AB, S4AB, S5AB, S6, S7, S8, S9, S10ABC, S11ABC, S12, S13AB, either as a supplementary data file or as a permanent DOI’d deposition.

d) Please cite the location of the data clearly in all relevant main and supplementary Figure legends, e.g. “The data underlying this Figure can be found in S1 Data” or “The data underlying this Figure can be found in https://doi.org/10.5281/zenodo.XXXXX”

e) Please make any custom code available, either as a supplementary file or as part of your data deposition.

f) Please ensure that your NCBI BioProject deposition is live and include the appropriate accession number in your manuscript and Data Availability Statement.

We expect to receive your revised manuscript within two weeks. 

*Published Peer Review History*

*Press*

Sincerely,

Roli Roberts

Roland Roberts PhD

Senior Editor

PLOS Biology

rroberts@plos.org

on behalf of

Richard Hodge, 

Senior Editor,

rhodge@plos.org,

PLOS Biology

DATA POLICY:

[Figs….]

CODE POLICY

Per journal policy, as the code that you have generated is important to support the conclusions of your manuscript, we require that you make it available without restrictions upon publication. Please ensure that the code is sufficiently well documented and reusable, and that your Data Statement in the Editorial Manager submission system accurately describes where your code can be found.

DATA NOT SHOWN?

COMMENTS FROM THE ACADEMIC EDITOR:

I think that the authors satisfactorily addressed the reviewer comments and the manuscript should soon be accepted. I did however find a large number of incorrect figure citations, so I ask that the authors check all their figure citations carefully, including all the supplementary figures which I did not check. I also found a few minor issues that I would like them to address:

Minor clarifications:

Muller C is sometimes referred to as muller C and sometimes as Neo X. Is this because for some analysis you do not distinguish between gametologs and in others you do? Clarification would be helpful.

Lines 150-151 add (Muller C) after Neo-X to remind the reader, and for consistency.

Lines 174-176 “We find that prior to meiosis (in GSC, SG, and SC), Muller C consistently has the lowest expression, with Muller AD showing intermediate expression levels between the two (Figure 2B).” When you say “between the two” in this sentence I think you mean between muller A and Muller C. But because you do not refer to Muller A directly it is confusing.

Figure citation problems:

Line 197, I think you mean to refer to Figure 2C, where the expression of MSL proximal and distal genes are shown.

Line 199, I think you mean to cite Figure 2C, not figure 2A

Figure 2D is not cited.

Line 207, I think you meant to cite Figure 2E.

Lines 272-274, is there a figure or supplementary figure that supports this claim?

Line 275, sentence should end with a period.

Line 291, do you mean to refer to the top a md middle panels specifically because Muller C is in the bottom panel

Line 518, I think you mean Figure 1F.

Line 537, I do not think you intended to cite Figure 3B here.

Line 539, do you mean 2B and 2C?

Line 545, I think you meant to cite figure 5?

Minor interpretation clarifications:

Line 373, I believe your statistical test shows that the neoY only genes are more highly expressed than NeoY biased genes and others. Note that they are overrepresented among highly expressed genes as is stated in the text.

Similarly on line 381, the figures show that Neo Y only genes have a strong testes bias in pseuoobscura, and not that they are overrepresented among genes with strong testes bias as the text states.

Lines 430-432, I am a little confused as to the mechanism of the “dosage dependent conflict over segregation”. While admittedly not my area of expertise, I think that segregation conflicts are thought mainly occur in meiotic drive mechanisms acting in the female germline, while meiotic drivers in male germlines act through a “poison” model in which one sperm class produces something rendering the alternative sperm class inviable. Can you refine your statement in light of what is known about how male meiotic drivers work?

Related to this point on line 571 “meiotic drivers that try to cheat fair meiosis” I find to be a more accurate description of female as compared to male meiotic drive.

Figure 7 I think it would be helpful to group the genes according to the expression/amplification patterns you describe in the text

---

## [Editor Report · Decision Letter 3]

27 Mar 2024

Dear Dr Bachtrog,

Thank you for the submission of your revised Research Article "Single cell RNA-seq of Drosophila miranda testis reveals the evolution and trajectory of germline sex chromosome regulation" for publication in PLOS Biology. On behalf of my colleagues and the Academic Editor, Erin Kelleher, I am pleased to say that we can accept your manuscript for publication, provided you address any remaining formatting and reporting issues. These will be detailed in an email you should receive within 2-3 business days from our colleagues in the journal operations team; no action is required from you until then. Please note that we will not be able to formally accept your manuscript and schedule it for publication until you have completed any requested changes.

PRESS

Best wishes, 

Richard

Richard Hodge, PhD

rhodge@plos.org

PLOS
